# Capacity of *Pseudomonas* Strains to Degrade Hydrocarbons, Produce Auxins and Maintain Plant Growth under Normal Conditions and in the Presence of Petroleum Contaminants

**DOI:** 10.3390/plants9030379

**Published:** 2020-03-19

**Authors:** Margarita Bakaeva, Elena Kuzina, Lidiya Vysotskaya, Guzel Kudoyarova, Tat’yana Arkhipova, Gulnaz Rafikova, Sergey Chetverikov, Tat’yana Korshunova, Dar’ya Chetverikova, Oleg Loginov

**Affiliations:** Ufa Institute of Biology, Ufa Federal Research Centre, Russian Academy of Sciences, Ufa 450054, Russia; margo22@yandex.ru (M.B.); misshalen@mail.ru (E.K.); vysotskaya@anrb.ru (L.V.); tnarkhipova@mail.ru (T.A.); rgf07@mail.ru (G.R.); che-kov@mail.ru (S.C.); lab.biotech@yandex.ru (T.K.); belka-strelka8031@yandex.ru (D.C.); biolab316@yandex.ru (O.L.)

**Keywords:** auxin, petroleum pollution, phytoremediation, plant growth promoting bacteria

## Abstract

The phytoremediation of soil contaminated with petroleum oil products relies on co-operation between plants and rhizosphere bacteria, including the plant growth-promoting effect of the bacteria. We studied the capacity of strains of *Pseudomonas*, selected as oil degraders, to produce plant hormones and promote plant growth. Strains with intermediate auxin production were the most effective in stimulating the seedling growth of seven plant species under normal conditions. Bacterial seed treatment resulted in about a 1.6-fold increase in the weight of barley seedlings, with the increment being much lower in other plant species. The strains *P. plecoglossicida* 2.4-D and *P. hunanensis* IB C7, characterized by highly efficient oil degradation (about 70%) and stable intermediate *in vitro* auxin production in the presence of oil, were selected for further study with barley. These strains increased the seed germination percentage approximately two-fold under 5% oil concentration in the soil, while a positive effect on further seedling growth was significant when the oil concentration was raised to 8%. This resulted in a 1.3–1.7-fold increase in the seedling mass after 7 days of growth, depending on the bacterial strain. Thus, strains of oil-degrading bacteria selected for their intermediate and stable production of auxin were found to be effective ameliorators of plant growth inhibition resulting from petroleum stress.

## 1. Introduction

Oil-producing and oil-refining industries result in the global contamination of the environment with hydrocarbons, which are toxic for plants, animals and humans [1,2,3]. Consequently, the goal of the development and application of effective approaches for the remediation of land contaminated with oil is of considerable current interest. Sowing wild and cultivate species on soils contaminated with oil is frequently used as a means of remediation [4,5,6]. The resulting plant population prevents soil erosion and helps to detoxify hydrocarbons and support higher numbers of hydrocarbon-oxidizing microorganisms. This effect has been ascribed to root exudates providing a substrate for bacterial growth [7,8,9,10]. The development of root systems also increases soil porosity, thereby elevating the mass transfer of respiratory substrates and electron acceptors [11]. Plants are tolerant of numerous environmental contaminants and are often used within phytoremediation strategies, but their biomass accumulation (and thereby the rate of contaminant removal) may be limited in the presence of high contamination [12]. Adding plant growth-promoting bacteria (PGPB) to soil can increase the accumulation of plant biomass in the presence of petroleum pollutants [13,14] and also increase plant tolerance to other stresses [15] such as drought [16,17], a deficiency of mineral nutrients [18], and presence of pesticides [19]. Thus, combining the capacity of petroleum destruction and the promotion of plant growth in the same bacterial culture is a promising approach to phytoremediation. 

Some reports suggest that certain plant hormones can ameliorate the negative effects of oil pollution on plants. Han et al. [20] showed that the plant hormone brassinolide could promote the photosynthesis of *Robinia pseudoacacia* seedlings in petroleum-stress conditions, thereby counteracting its adverse effects. Microorganisms can directly influence plant growth by synthesizing growth-stimulating hormones [21,22,23]. Nevertheless, only a small number of publications have addressed the synthesis of hormones by plant growth promoting rhizobacteria (PGPR) under conditions of oil pollution and their phytoremedial effect on plants. For example, a model system consisting of bacteria (*Sinorhizobium meliloti*) capable of associative nitrogen fixation, hydrocarbon oxidation, and the production of auxins in combination with a grass (*Sorghum bicolor*) has been successfully used for the remediation of soils contaminated with polycyclic aromatic hydrocarbons [24]. Testing free living bacteria of *Pseudomonas* as hormone producers and plant growth promoters under conditions of contamination with petroleum is of no less interest than the study of associative bacteria, since the capacity for the active decomposition of hydrocarbons and resistance to adverse environmental conditions have been detected in many representatives of this genus [25,26,27].

The goal of the present work was to study the ability of strains of *Pseudomonas* bacteria known to be effective in breaking-down oil hydrocarbons to synthesize auxins and to follow their effects on the growth of plants in normal conditions and under petroleum stress. We have chosen the following plant species which have been reported to be relatively resistant to petroleum pollutants to study the effects of seed inoculation on germination and seedling growth under normal conditions and in the presence of oil or individual hydrocarbons: oats (*Avena sativa* L. [4]), barley (*Hordeum vulgare* L. [28]), Sudan grass (*Sorghum × drummondii* [29]), pea (*Pisum sativum* L. [30]), smooth brome (*Bromus inermis* Leyss [31]), meadow fescue (*Festuca pratensis* Huds. [32]), and clover (*Trifolium pratense* L. [33]).

## 2. Results and Discussion

### 2.1. Properties of Microorganisms

Six strains belonging to different species of *Pseudomonas* were selected from a collection at the Ufa Institute of Biology, Ufa Federal Research Centre, Russian Academy of Sciences (UIB UFRC RAS) as being capable of oxidizing oil products and accumulating indole-3-acetic acid (IAA) in culture media. Their characteristics are presented in Table 1 and Table 2. A description of the *Pseudomonas turukhanskensis* IB 1.1 strain was reported by Korshunova et al. [25], while the *P. plecoglossicida* 2.4-D strain was described in the article by Chetverikov et al. [34].

The presence of petroleum hydrocarbons in the soil is known to make mineral nutrients less accessible for plants [35]. Hydrocarbon-oxidizing bacteria also have an increased need for sources of nitrogen and phosphorus, since they are almost absent from oil products. Therefore, the ability of bacteria to fix atmospheric nitrogen and transfer phosphorus salts into a soluble form could have a positive effect on the phytomelioration of contaminated soils. Among the studied strains, those of *P. hunanensis* IB C7, *P. nitroreducens* IB ND 1.1, and *P. turukhanskensis* IB 1.1 are nitrogen-fixers (diazotrophy) and were found capable of mobilizing water-insoluble phosphates (Figure 1). Nitrogenase activity was significantly higher than in the control (about 14, 13 and 18 mmol of C_2_H_4_ h ^−1^ ml ^−1^, respectively, with this level being considered rather low). Around the colonies of the *P. turukhanskensis* IB 1.1 strain, phosphates were dissolved on Pikovskaya medium in a zone with a radius of 12 mm; for *P. hunanensis* IB C7 strain, the radius was 4 mm; and for *P. nitroreducens* IB ND, the radius was 1.1–3 mm. In the remaining studied strains, phosphate dissolution was not visually detected, and the values of nitrogenase activity did not differ significantly from the control sample.

To clarify the phylogenetic position of the selected strains and their relationship with each other and with other species of *Pseudomonas*, a phylogenetic tree was constructed based on the nucleotide sequence of the 16S rRNA gene (Figure 2). The dendrogram shows that the strains belong to different clusters.

Table 2 shows that the strains of *Pseudomonas* we selected could breakdown crude oil, diesel fuel, marine fuel oil, lubricating oil and heavy fuel oil. *P. plecoglossicida* 2.4-D, *P. hunanensis* IB C7, *P. nitroreducens* IB ND 1.1 and *P. turukhanskensis* IB 1.1 actively destroyed both crude oil and its products, while *P. extremaustralis* IB K2 and *P. sihuiensis* IB P1 were less active, although *P. extremaustralis* IB K2 was almost as active as *P. hunanensis* IB C7 in metabolizing heavy fuel oil.

The concentration of the auxin indolyl-3-acetic acid (IAA) in the nutrient medium reached a maximum when cultures were transferred to the stationary phase of growth [18]. At this stage, *Pseudomonas* spp. significantly differed from each other in their ability to accumulate IAA in liquid culture (Table 3). On this basis, the strains could be divided into groups with high (*P. turukhanskensis* IB 1.1), intermediate (*P. plecoglossicida* 2.4-D, *P. hunanensis* IB C7, *P. nitroreducens* IB ND 1.1, *P. sihuiensis* IB P1) or low IAA production (*P. extremaustralis* IB K2). Cytokinins and abscisic acid were present in much lower concentrations than IAA in the culture media of 0.4–2.8 ng mL^−1^ and 0.4–1.2 ng mL^−1^, respectively.

Auxin concentration depended on the presence of oil hydrocarbons in the culture media and varied from about 300 ng L^−1^ to more than 11000 ng mL^−1^, when the bacteria grew on contaminant-free medium. Crude oil decreased these values to between 120 ng mL^−1^ and 6048 ng mL^−1^, while the range in hexadecane was 84 ng mL^−1^ to 7301 ng mL^−1^ and the range in toluene was 83 ng mL^−1^ to 140 ng mL^−1^ (Table 3). Thus, the presence of crude oil and its refined products in the culture media decreased the accumulation of IAA. This was probably a result of the inhibition of bacterial growth by these substances (see Table 4). Among the studied strains, *P. hunanensis* IB C7 was particularly noteworthy, since no significant decrease was found in IAA accumulation in the presence of oil or hexadecane. Toluene had a strong toxic effect on all the cultures, inhibiting both their reproduction (Table 4) and the synthesis of IAA (Table 3).

A weak overall correlation (*r* = 0.5) was found between the decline in IAA accumulation in the culture medium under the influence of oil and the rate of oil destruction by bacteria of individual strains. It is likely that ability of *P. hunanensis* IB C7 to synthesize IAA in amounts unaffected by oil or hexadecane is not related directly with its oil breakdown activity.

### 2.2. Effect of Microorganisms on Plants 

The seven plant species used were selected on the basis of being known as somewhat resistant to oil contamination of soil: oats (*Avena sativa* L. [4]), barley (*Hordeum vulgare* L. [28]), Sudan grass (*Sorghum × drummondii* [29]), pea (*Pisum sativum* L. [30]), smooth brome (*Bromus inermis* Leyss [31]), meadow fescue (*Festuca pratensis* Huds. [32]), and clover (*Trifolium pratense* L. [33]). However, it was important to test how their performance was influenced by the selected hydrocarbon-degrading bacteria. Table 5 and Table 6 show that the species differed in their growth response to seed inoculation with different bacterial strains. An increase in germination percentage under Petri dish conditions was detected in *Avena sativa*, *Hordeum vulgare*, *Pisum sativum*, *Bromus inermis*, *Trifolium pratense*, while germination of *Sorghum × drummondii* and *Festuca pratensis* was not changed significantly by any of the six bacterial strains tested (Table 5). The seedling mass of *Trifolium pratense, Hordeum vulgare, Pisum sativum, Avena sativa, Festuca pratensis* and *Sorghum × drummondii* after 7 days from imbibition was increased by bacterial treatment but had no promoting effect on *Bromus inermis* (Table 6). Growth promotion was mostly detected when seeds were inoculated with bacterial strains characterized by an intermediate accumulation of auxin in the culture medium. Thus, seed treatment with *P. hunanensis* IB C7 strain increased germination in three plant species and increased the seedling mass in five out of seven species. Similarly, *P. nitroreducens* IB ND 1.1 elevated germination in two species and seedling mass in five species while *P. plecoglossicida* 2.4-D did so in four and two species and *P. sihuiensis* IB P1 improved germination a subsequent seedling growth in two and three species, respectively. *P. extremaustralis* IB K2, characterized by relatively low levels of IAA accumulation in the culture media, did not influence seed germination in any of the species tested and stimulated mass accumulation in only two species, while *P. turukhanskensis* IB 1.1 (a strain generating extremely high levels of auxin in the culture medium) stimulated germination and mass accumulation less than bacterial strains with intermediate level of auxin accumulation (in two and one plant species, correspondingly (Table 6).

The absence of plant growth promotion by low-auxin bacteria confirms the importance of this hormone for the bacterial effect on plant growth (Spaepen and Vanderleyden [21] and references therein). The lack of a stimulating effect in control conditions by the strain with the highest level of auxin production (*P. nitroreducens* IB ND 1.1.) is in accordance with the ability of a high concentration of IAA to stimulate ethylene production capable of inhibiting plant growth [36]. 

For further experiments, we chose *P. hunanensis* IB C7 and *P. plecoglossicida* 2.4-D strains characterized by an active capacity for degrading hydrocarbons and for promoting seedling growth (Table 2) and with an intermediate but relatively stable level of IAA production (Table 3). These strains were used to inoculate barley seeds sown in soil. Barley was chosen as it proved to be the most responsive to bacterial treatment in Petri dish experiments in the absence of oil (Table 5 and Table 6). The presence of oil-polluted soil was found to decrease germination and the growth of barley seedlings untreated with bacteria (Figure 3). The toxic effects of oil pollutants are known to be most important at the early stages of plant growth [29]; therefore, it was important to check whether bacterial inoculation can also stimulate germination and growth in petroleum-stressed-seedlings.

Inoculation with *P. plecoglossicida* 2.4-D increased the germination of barley in the absence of oil pollutants and at both 5% and 8% oil treatments. Treatment with *P. hunanensis* IB C7 promoted germination only at the lower oil concentration (5%) (or without oil pollutant). Thus, the inhibition of germination from oil contamination was diminished by both bacterial strains at 5% oil contamination but only by *P. plecoglossicida* 2.4-D at 8% oil concentration. These findings agree with the Petri dish experiments in Petri dishes, with seedling mass being increased by both bacterial strains in barley grown in containers with soil without oil pollutants. At the lower (5%) oil concentration in containers of soil, plants treated with the bacterial strains did not significantly differ from the untreated plants in terms of seedling mass and shoot length. At this level of contamination, *P. hunanensis* IB C7 increased the total length of barley primary roots. The response to higher oil concentration was different. In this case, although soil contamination with oil dramatically decreased plant growth, significant increases in seedling mass or shoot and root length were detected in plants treated with either bacterial strain. Plant growth maintenance under a high level of pollution is likely to result from either the decreased toxicity of pollutants resulting from their active destruction by bacteria or from the direct effect of bacterial hormones.

It is of interest that no nitrogenase or phosphate-mobilizing activity was found in the *P. plecoglossicida* 2,4-D strain, while the treatment of plants with this strain stimulated their growth against the background of oil pollution. The growth-stimulating activity of these bacteria could perhaps be explained by their ability to synthesize auxins. These results emphasize the importance of the production of hormones by bacteria and their action under stressful conditions. Plants are usually used for the remediation of polluted soil at the late stages of the process, when the concentration of pollutants is already lowered. Table 6 shows that the germination and growth of plants themselves (without bacteria treatment) greatly decreased with an increase in oil concentration. However, seed inoculation with appropriate bacterial strains diminished the extent of growth inhibition caused by increased oil contamination. These results suggest that an inoculation such as this may strengthen the ability of plants to remediate highly contaminated soils.

Thus, strains of bacteria that can more readily metabolize oil or oil-derived products exert a promotive action on germination and mass accumulation by *Trifolium pratense, Hordeum vulgare, Pisum sativum, Avena sativa*. Such bacterial strains differ from each other in their capacity to accumulate auxins in their culture media, and this capacity decreases in the presence of oil. The stimulating effects of inoculating seeds on plant germination and growth are more clearly pronounced in bacteria with an intermediate and stable level of auxin production. Treating barley seeds with these bacterial strains of *Pseudomonas* in particular was found to increase germination and root growth in soil contaminated with petroleum oil during the first week of germination.

## 3. Materials and Methods 

We studied bacterial strains recovered by the authors of the present report from soil and water contaminated with oil and stored in the culture collections of the Ufa Institute of Biology of the Russian Academy of Sciences (UIB, RAS) (Table 1). For the taxonomic affiliation of bacterial strains, 16S rRNA gene nucleotide sequences were determined. Total DNA from bacterial colonies was recovered with the reagent kit «RIBO-sorb» (Amplisens^®^, Central Research Institute of Epidemiology, Moscow, Russian Federation) according to the recommendations of the manufacturer. The amplification of the fragment of the 16S rRNA gene was carried out using universal primers: 27F (5′-AGAGTTTGATCTGGCTCAG-3′) and 1492R (5′-ACGGTACCTTGTTACGACTT-3′) [37]. Sequencing of 16S rRNA was performed using BigDye Terminator Sequencing Kit (Applied Biosystems, Thermo Fisher Scientific Inc., Waltham, MA, USA) with the help of the Genetic Analyzer 3500 xL (Applied Biosystems, Thermo Fisher Scientific Inc., Waltham, MA, USA). To sequester cycle-sequencing reaction components, we used the BigDye^®^ XTerminator™ Purification Kit (Applied Biosystems, Thermo Fisher Scientific Inc., Waltham, MA, USA). The search for 16S rRNA nucleotide sequences similar to the corresponding sequences of the studied strains was carried out using the GenBank sequence database with the help of the software package BLAST (http://www.ncbi.nlm.nih.gov/blast).

The capacity of bacteria for the oxidation of petroleum hydrocarbons was qualitatively evaluated by measuring their growth on Raymond mineral culture medium (g L^−1^)—NH_4_NO_3_, 2.0; MgSO_4_x7H_2_O, 0.2; KH_2_PO_4_, 2.0; Na_2_HPO_4_, 3; CaCl_2_x6H2O, 0.01; Na_2_CO_3_, 0.1; pH, 7.0 [38]—containing 2 g L^-1^ of oil products as the only source of carbon. Quantitative evaluation was carried out by measuring the decline in oil products in Raymond mineral culture medium during the growth of pure bacterial cultures. The following hydrocarbons were used to prepare the culture media: oil (brand Urals), heavy fuel oil, marine fuel oil, diesel fuel, lubricating oil. Concentrations of hydrocarbons in the samples was measured by the Luminescent/Photometric Liquid Analyzer “Fluorat-02” (“Lumex-Marketing”, St.-Petersburg, Russia).

To study the capacity of bacteria to produce plant hormones, bacteria were grown on King B liquid nutrient medium (g L^−1^)—peptone, 20.0; glycerol, 10.0; K_2_HPO_4_, 1.5; MgSO_4_x7H_2_, 1.5 [39]—to which hydrocarbons were added to yield final concentration of 1 g L^−1^. The transfer of bacterial suspensions of test strains prepared with sterile tap water into the nutrient medium was performed to yield its final titer (1 ± 0.5) × 10^6^ CFU mL^−1^. Microorganisms were cultivated in Erlenmeyer flasks on a thermostatically controlled shaker (160 rotation per minute) at 28 °C for 72–168 h.

The capacity of *Pseudomonas* strains to synthesize auxin, IAA, was measured by immunoenzyme assay. Before the extraction of IAA, culture media were centrifuged at 10000 rpm for 10 min. Aliquots of supernatant were partitioned with diethyl ether according to a modified scheme with a diminished volume of extractant used for each stage of extraction and re-extraction [40]. Abscisic acid (ABA) and cytokinin purification and assay was carried out as described [41]. An enzyme-linked immunosorbent assay was carried out as described [18,41]. 

The ability to mobilize phosphates was evaluated by measuring the size of transparent zones on Pikovskaya medium [42]. The acetylene reduction assay was used as a measure of bacterial nitrogenase activity, with ethylene used as a marker and measured by gas chromatography [43].

Effect of Pseudomonas strains on plants was studied using seedlings of oats (*Avena sativa* L., cv. Konkur), barley (*Hordeum vulgare* L., cv. Chelyabinskiy), Sudan grass (*Sorghum × drummondii*, cv. Chishminskaya early), pea (*Pisum sativum* L., cv. Chishminskaya 229), smooth brome (*Bromus inermis* Leyss, cv. Chishminskaya 3), meadow fescue (*Festuca pratensis* Huds., cv. Ufimka) and clover (*Trifolium pratense* L., cv. Early 2). Preparations for seed inoculation were obtained by cultivating bacteria for 3 days in the liquid King B medium. Seed inoculation was carried out by soaking in bacterial suspension (10^5^ bacterial cells per seed) for an hour-s. Control seeds were similarly soaked in water. Seeds were placed in Petri dishes (20 seeds per dish in 10 replicates for each variant) on moistened filter paper or sowed in containers with 60 g of soil and incubated for 7 days at 20 °C under illumination about 100 μmol m^−2^s^−1^. The soil was artificially polluted with oil in concentrations of 50 and 80 g kg^−1^. Humidity was maintained at 80%. Germination percentage, seedling mass, shoot length and total length of primary roots were measured 7 days after seed soaking.

### Statistics

Data were expressed as means ± standard error, which were calculated in all treatments using MS Excel. Significant differences between means were analyzed by a *t*-test.

## Figures and Tables

**Figure 1 plants-09-00379-f001:**
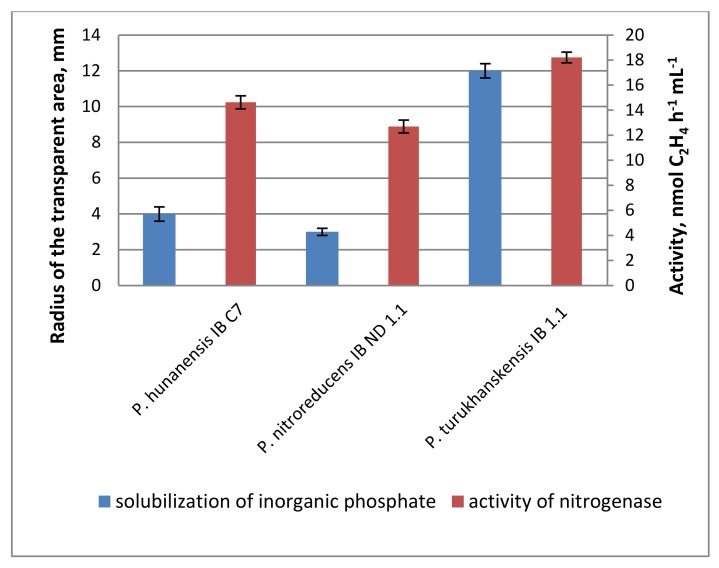
Solubilization of inorganic phosphate and activity of nitrogenase in liquid bacterial cultures of strains *P. hunanensis* IB C7, *P. nitroreducens* IB ND 1.1, and *P. turukhanskensis* IB 1.1. Mean values ± SE are presented (*n* = 10 plates for phosphate solubilization, *n* = four flasks investigating the activity of nitrogenase).

**Figure 2 plants-09-00379-f002:**
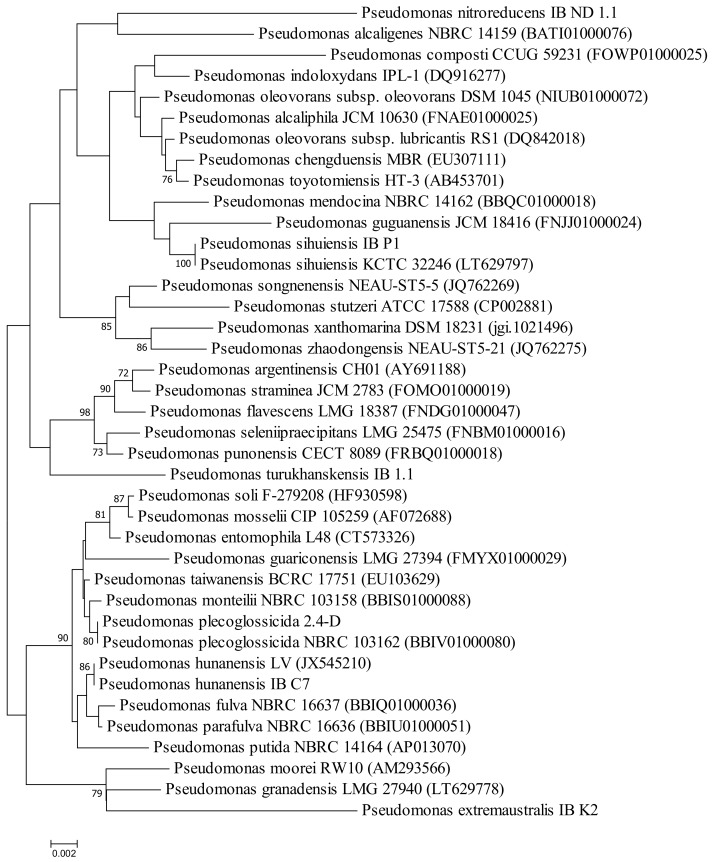
Neighbor-joining phylogenetic tree based on 16S rRNA gene sequences of the selected hydrocarbon-degrading bacteria and closely related species of the genus *Pseudomonas*. Bootstrap values (expressed as percentages of 1000 replications) are shown at the branching points. Bar—two nucleotide substitutions per 100 nt.

**Figure 3 plants-09-00379-f003:**
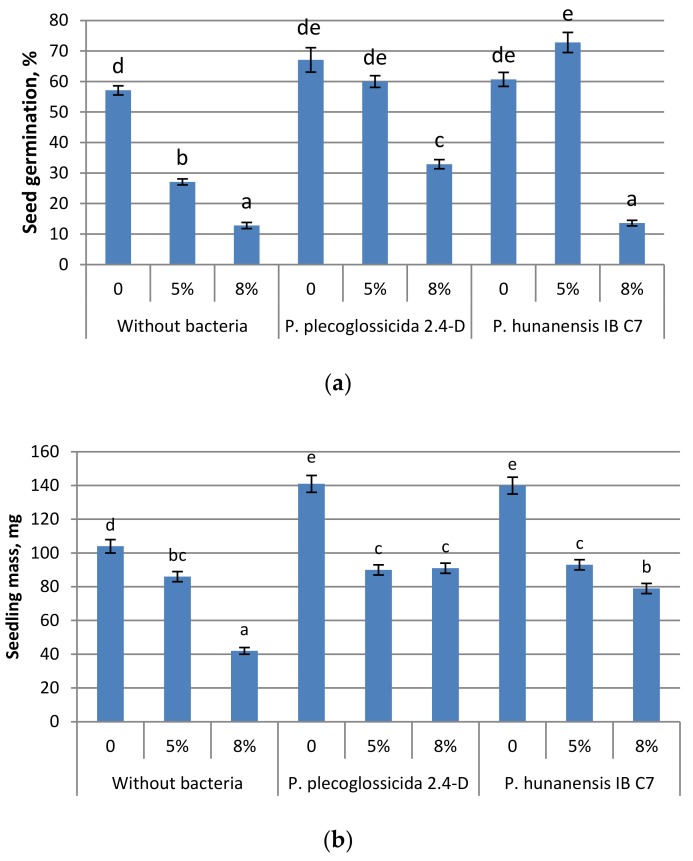
Effect of oil pollutants (50 and 80 g kg^−1^ soil) and inoculation of seeds with strains of *Pseudomonas* bacteria on (**a**) germination, (**b**) seedling mass and (**c**) length of barley seedlings grown in soil and measured 7 days after seed soaking. Mean values ± SE are presented (*n* = 30). Significantly different means of each parameter are marked with different letters (*p* ≤ 0.05, *t*-test).

**Table 1 plants-09-00379-t001:** Species affiliation and origin of bacterial strains used in the experiments.

Strain Number in the Collection of UIB UFRC RAS	Species/Strain	16S rRNA Nucleotide Sequence Similarity, %	Strain Origin
IB K2	*Pseudomonas extremaustralis* 14-3^T^	99.93	Soil contaminated with oil (Republic of Bashkortostan, Russia)
IB C7 (B-3229D)	*P. hunanensis* LV^T^	99.56	Steppe soil (Sol-Iletsky District, Orenburg region, Russia)
IB ND 1.1	*P. nitroreducens* DSM 14399	99.90	Soil from the site of oil spill (Khanty-Mansi Autonomous Area, Russia)
IB P1	*P. sihuiensis* KCTC 32246^T^	99.06	Bilge water from a transport vessel (port of Novorossiysk, Russia)

**Table 2 plants-09-00379-t002:** Degradation of oil and oil products by strains of *Pseudomonas* after 3 days of cultivation on Raymond media containing 2 g L^−1^ of hydrocarbon substrate.

Strains	Extent of Degradation of Hydrocarbon Substrate, %				
	Oil	Diesel Fuel	Marine Fuel Oil	Lubricating Oil	Heavy Fuel Oil
*Pseudomonas extremaustralis* IB K2	51.3 ± 0.7^a^	65.5 ± 3.4 ^a^	47.6 ± 1.3 ^a^	17.1 ± 0.3 ^c^	22.2 ± 0.3 ^d^
*P. hunanensis* IB C7	73.0 ± 1.6 ^b^	87.9 ± 2.2 ^c^	59.0 ± 1.2 ^c^	34.7 ± 0.4 ^d^	24.8 ± 0.2 ^e^
*P. nitroreducens* IB ND 1.1	83.3 ± 1.6 ^c^	90.2 ± 1.8 ^c^	45.5 ± 0.9 ^a^	12.0 ± 0.3 ^a^	15.2 ± 0.4 ^b^
*P. plecoglossicida* 2.4-D	69.0 ± 1.5 ^b^	80.5 ± 2.0 ^b^	Nm	Nm	19.7 ± 0.3 ^c^
*P. sihuiensis* IB P1	54.0 ± 1.9 ^a^	60.7 ± 3.2 ^a^	47.9 ± 1.1 ^a^	13.0 ± 0.2 ^b^	13.8 ± 0.4 ^a^
*P. turukhanskensis* IB 1.1	84.5 ± 2.1 ^c^	89.0 ± 3.0 ^c^	51.6 ± 1.1 ^b^	18.0 ± 0.5 ^c^	20.5 ± 0.5 ^c^

Nm—not measured. Mean values ± SE are presented (*n* = 10). In each column, significantly different means are marked with different letters (*p* ≤ 0.05, *t*-test).

**Table 3 plants-09-00379-t003:** Auxin production (ng mL^−1^) by bacterial strains cultivated on King B medium with different additives: oil, hexadecane and toluene (2 g L^−1^)^.^

Strains	Without Additives	Oil	Hexadecane	Toluene
*Pseudomonas extremaustralis* IB K2	318 ± 37 ^c^	120 ± 13 ^ab^	84 ± 3 ^a^	140 ± 21 ^ab^
*P. hunanensis* IB C7	661 ± 88 ^d^	702 ± 25 ^d^	654 ± 25 ^d^	80 ± 5 ^a^
*P. nitroreducens* IB ND 1.1	1413 ± 64 ^e^	289 ± 40 ^c^	692 ± 94 ^d^	102 ± 10 ^a^
*P. plecoglossicida* 2.4-D	1205 ± 72 ^e^	588 ± 32 ^d^	518 ± 21 ^d^	89 ± 4 ^a^
*P. sihuiensis* IB P1	1324 ± 38 ^e^	189 ± 36 ^bc^	698 ± 7 ^d^	83 ± 4 ^a^
*P. turukhanskensis* IB 1.1	11261 ± 897 ^g^	6048 ± 809 ^f^	7301 ± 406 ^f^	154 ± 12 ^b^

Mean values ± SD are presented (*n* = 12). Significantly different means are marked with different letters (*p* ≤ 0.05, *t*-test).

**Table 4 plants-09-00379-t004:** The effect of oil, hexadecane and toluene (2 g L^−1^) on number of colonies forming units (CFU mL^−1^) in culture media during the stationary phase of their growth on the King B medium.

Strains	Additives to the Medium			
	Without Additives	+ Oil	+ Hexadecane	+ Toluene
*Pseudomonas extremaustralis* IB K2	(5.1 ± 0.2) × 10^9^	(1.8 ± 0.1) × 10^8^	(2.0 ± 0.1) × 10^8^	(2.0 ± 0.1) × 10^5^
*P. hunanensis* IB C7	(7.9 ± 0.2) × 10^9^	(8.0 ± 0.2) × 10^8^	(1.0 ± 0.1) × 10^9^	(3.0 ± 0.1) × 10^5^
*P. nitroreducens* IB ND 1.1	(5.7 ± 0.3) × 10^9^	(2.6 ± 0.1) × 10^8^	(2.3 ± 0.1) × 10^8^	<10^5^
*P. plecoglossicida* 2.4-D	(5.4 ± 0.3) × 10^9^	(2.0 ± 0.1) × 10^8^	(2.6 ± 0.1) × 10^8^	<10^5^
*P. sihuiensis* IB P1	(1.0 ± 0.1) × 10^10^	(9.0 ± 0.2) × 10^7^	(8.0 ± 0.2) × 10^7^	<10^5^
*P. turukhanskensis* IB 1.1	(4.0 ± 0.2) × ·10^9^	(7.4 ± 0.2) × 10^8^	(3.3 ± 0.2) × 10^8^	<10^5^

Mean values ± SE are presented (*n* = 12).

**Table 5 plants-09-00379-t005:** Effect of inoculating seeds with strains of *Pseudomonas* bacteria on percentage seed germination in Petri dishes after 7 days.

Plant Species	*Pseudomonas extremaustralis* IB K2	*P. hunanensis* IB C7	*P. nitroreducens* IB ND 1.1	*P. plecoglossicida* 2.4-D	*P. sihuiensis* IB P1	*P. turukhanskensis* IB 1.1	Control
*Avena sativa*	34.40 ± 1.33	39.00 ± 1.15	**56.10** ± 2.14	**52.50** ± 1.82	36.60 ± 1.50	37.50 ± 1.26	40.00 ± 2.05
*Bromus inermis*	57.90 ± 2.30	**73.33** ± 1.04	66.05 ± 2.17	**69.38** ± 1.76	56.65 ± 1.22	59.38 ± 1.08	59.18 ± 1.40
*Festuca pratensis*	40.62 ± 1.85	44.38 ± 1.35	37.00 ± 1.28	39.38 ± 1.15	43.75 ± 2.18	40.63 ± 1.43	43.13 ± 1.07
*Hordeum vulgare*	96.65 ± 2.5	96.65 ± 2.42	96.65 ± 1.98	91.65 ± 2.13	**97.50** ± 1.63	**97.50** ± 1.54	91.65 ± 2.00
*Pisum sativum*	95.00 ± 2.73	**96.65** ± 1.84	90.00 ± 2.87	**98.35** ± 2.15	**98.35** ± 2.35	95.00 ± 2.46	91.10 ± 1.55
*Sorghum × drummondii*	80.00 ± 3.12	83.35 ± 2.25	88.35 ± 2.08	82.50 ± 1.81	78.35 ± 3.07	83.35 ± 2.85	83.35 ± 2.54
*Trifolium pratense*	39.18 ± 1.50	**60.00** ± 1.49	**69.18** ± 1.88	**60.83** ± 1.43	36.88 ± 1.40	**60.00** ± 1.80	35.00 ± 1.39

Mean values ± SE are presented (*n* = 10). Mean values for treated plants significantly different from the control (untreated plants, *p* ≤ 0.05, *t*-test) are marked with bold letters.

**Table 6 plants-09-00379-t006:** Effect of inoculating seeds with strains of *Pseudomonas* bacteria on the mass of seedlings of plant species grown in Petri dishes in the absence of oil, hexadecane or toluene.

Plant Species	*Pseudomonas extremaustralis* IB K2	*P. hunanensis* IB C7	*P. nitroreducens* IB ND 1.1	*P. plecoglossicida* 2.4-D	*P. sihuiensis* IB P1	*P. turukhanskensis* IB 1.1	Control
*Avena sativa*	**86.0** ± 1.2	**112.4** ± 2.3	**80.5** ± 0.9	**99.3** ± 1.6	75.3 ± 1.1	74.6 ± 1.8	69.0 ± 1.5
*Bromus inermis*	21.5 ± 0.41	20.6 ± 0.18	17.7 ± 0.58	20.2 ± 0.43	17.3 ± 0.53	23.7 ± 0.61	19.5 ± 0.29
*Festuca pratensis*	3.90 ± 0.18	**4.88** ± 0.07	4.23 ± 0.09	3.88 ± 0.24	**4.72** ± 0.08	**4.84** ± 0.09	4.06 ± 0.10
*Hordeum vulgare*	**150** ± 3	**172** ± 5	**160** ± 3	**156** ± 4	**179** ± 5	**179** ± 5	105 ± 2
*Pisum sativum*	626 ± 9	**643** ± 4	**702** ± 10	591 ± 14	**669** ± 4	602 ± 5	598 ± 4
*Sorghum × drummondii*	28.0 ± 0.4	40.2 ± 0.9	37.7 ± 0.6	**44.1** ± 0.6	39 ± 0.7	41.3 ± 1.0	34 ± 0.5
*Trifolium pratense*	2.67 ± 0.27	**6.05** ± 0.25	**4.53** ± 0.23	2.54 ± 0.04	2.84 ± 0.14	2.76 ± 0.17	2.60 ± 0.25

Comments: Mean values ± SE are presented (*n* = 40). Mean values for treated plants significantly different from the control (untreated plants, *p* ≤ 0.05, *t*-test) are marked with bold letters.

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
