# Peer review of "Capacity of Pseudomonas Strains to Degrade Hydrocarbons, Produce Auxins and Maintain Plant Growth under Normal Conditions and in the Presence of Petroleum Contaminants"

_plants, 2020, doi:10.3390/plants9030379_

Round 1
Reviewer 1 Report
The experimentation described in the paper: "Capacity of strains of Pseudomonas genus to destroy hydrocarbons, produces auxins and maintain plant growth under normal conditions and in the presence of petroleum contaminants" concerns the study of the effect of some hydrocarbon-oxidizing battery strains of the genus Pseudomonas on the growth of plants in presence of contamination from oil derivatives.
Before going into scientific details, I would like to emphasize that a significant revision of English is certainly necessary, especially in the construction of the sentences and in the use of some obsolete terms. It is often not easy to understand what the authors want to express.
Here are some specific comments.
Abstract
The only numbers concern the percentages of oil (5 and 8%) used in the experiments and the corresponding results are only described in the text. Please enter the data relating to the percentage of degradation and the growth of plants (tab. 7).
Introduction
lane 35: "detoxification of carbohydrates .... .high number of carbohydrate oxidizing microorganisms ..."?; Carbohydrates are not toxic, was it meant hydrocarbons?
Results and Discussion
Lane 68: six stains, correct with strains. In other parts of the document, this word must also be corrected.
Lane 76, Table 1: the table shows only 4 of the 6 strains mentioned
Lane 77, Table 2: was the degradation of the different oils calculated after only 3 days?
Lane 95: in the description of the results of table 3 also describe the results obtained with toluene
Lane 111: "bacteria-destructors", better the selected hydrocarbon-degrading bacteria
Lane 123: P. extramaustralis is not in italics
Lane 136-142: the importance of the bacterial auxin IAA is affirmed because it highlights the negative effect in the presence of low levels of IAA and also negative effect detected in the presence of high levels of IAA (caused by the stimulation of ethylene production), but all the molecules with plant growth-promoting properties produced by microorganisms are not taken into account. It would be appropriate to complete the study by also evaluating these (ACC deaminase, production of siderophores, ammonia, biofilm formation, nitrogen fixation, solubilization of inorganic phosphate ...).
Lane 152, Tab. 7: it would be much better to put these results in a bar graph, certainly of greater impact.
Materials and Methods
Lane 187: stains, correct with strains
Lane 191-201: if the strains come from a collection they have already been classified. Why is the sequencing of 16SrRNA fragments described?
Lane 217: for the qualitative and quantitative determination of the IAA, Salkowski's method is very rapid without the use of solvents.
Final tip:
the use of real soils (especially with not recent contaminations) would lead to much more solid results from a scientific point of view. The artificial contamination of the matrices (soils and/or waters) is often ineffective because in this way those microorganisms selected by nature to survive in those conditions are not evaluated.
The best way to study biodegradation is precisely the isolation, selection and characterization of these autochthonous microorganisms which can then be used for laboratory tests on their effectiveness.
Reviewer 2 Report
In the present study Bakaeva et al., report on Pseudomonas strains that are able to degrade oil and oil products. They give report in strains that help to increase plant germination rates and seedling biomass in the presence of oil. Furthermore, the auxin production by the strains was measured in the presence of oil. Even though, the topic is of interest for the readership of Plants, I am afraid that it needs some revision before it is publishable.
I am missing the Statistics part in the Materials and Methods section. Please add information on the programme and tests you used. Furthermore, description in Table 7 is not clear to me. "Comments: Mean values ± SE are presented (n=30). Significantly different means are marked with different letters (p≤0.05, t-test)." A t-test usually compares the means of only two groups. Why the authors chose the t-test?
Korshunova et al, 2016 presented a phylogenetic tree based on the 16S rRNA sequence for Pseudomonas turukhanskensissp. nov. IB1.1. The quality of the present manuscript would be enhanced by presenting phylogenetic trees for all strains used.
I would recommend using the MDPI English Editing Service or sending the manuscript to a native speaker for revision.
Minor:
Line 18: 7 -> seven
Line 68/76/Table 2/187: stains -> strains
Line 71: genus -> genus
Line 93 and others: ng/ml -> ng mL-1
Line 123 and others: P. extremaustralis -> P. extremaustralis
Line 218: rot/min -> SI ?
Line 233: wt.% -> SI ?
References need to be revised, e.g. ref. 20,21,22,27.
Round 2
Reviewer 1 Report
Comments to the revised version:
I agree with the changes made, thank you.
I have only a few comments:
- I had not suggested changes to the title, and I sincerely prefer the one of the first version or even the current one by making a small change: "Capacity of Pseudomonas strains to degrade ...." instead of "Capacity of strains of Pseudomonas bacteria to degrade ..."
- It seems to me that in the first sentence of the abstract, the verb is missing.
- About point 15 concerning my suggestion on the use of real soils, I accept your explanations. Still, I add that a molecular analysis with new generation sequencing (NGS systems) can undoubtedly solve the problem above you say about competition with species indigenous. The qualitative-quantitative report that this system generates can monitor in detail the progress of the entire microbial community present.
